# Intrusion Detection System for IoT: Analysis of PSD Robustness

**DOI:** 10.3390/s23042353

**Published:** 2023-02-20

**Authors:** Lamoussa Sanogo, Eric Alata, Alexandru Takacs, Daniela Dragomirescu

**Affiliations:** 1Laboratoire d’Analyse et d’Architecture des Systèmes du Centre National de la Recherche Scientifique (LAAS-CNRS), 31077 Toulouse, France; 2Institut National des Sciences Appliquées de Toulouse (INSA-Toulouse), 31400 Toulouse, France; 3Faculté Sciences et Ingénierie, Université Toulouse III–Paul Sabatier, 31062 Toulouse, France

**Keywords:** internet of things (IoT), device fingerprinting, relevant fingerprinting feature, power spectral density (PSD), device identification

## Abstract

The security of internet of things (IoT) devices remains a major concern. These devices are very vulnerable because of some of their particularities (limited in both their memory and computing power, and available energy) that make it impossible to implement traditional security mechanisms. Consequently, researchers are looking for new security mechanisms adapted to these devices and the networks of which they are part. One of the most promising new approaches is fingerprinting, which aims to identify a given device by associating it with a unique signature built from its unique intrinsic characteristics, i.e., inherent imperfections, introduced by the manufacturing processes of its hardware. However, according to state-of-the-art studies, the main challenge that fingerprinting faces is the nonrelevance of the fingerprinting features extracted from hardware imperfections. Since these hardware imperfections can reflect on the RF signal for a wireless communicating device, in this study, we aim to investigate whether or not the power spectral density (PSD) of a device’s RF signal could be a relevant feature for its fingerprinting, knowing that a relevant fingerprinting feature should remain stable regardless of the environmental conditions, over time and under influence of any other parameters. Through experiments, we were able to identify limits and possibilities of power spectral density (PSD) as a fingerprinting feature.

## 1. Introduction

Traditional computing devices, e.g., computers, have sufficient memory resources and high computing power, and their energy consumption is not often a concern as they are mostly plugged into the mains. Thanks to these characteristics, these traditional devices support traditional security mechanisms such as cryptographic algorithms, firewalls, virtual private networks (VPN), antivirus software, an intrusion detection system (IDS), and so on. These mechanisms make attacks more difficult and, thus, provide a certain level of security to these traditional computing devices.

In recent years, in the context of the internet of things (IoT), we have witnessed a proliferation of new types of computing devices that we will, henceforth, call “IoT devices”, e.g., internet-connected heart monitors. Unlike traditional devices, these IoT devices have both limited memory resources and limited computing power, and also require low energy consumption because they mostly use a power supply of limited capacity, e.g., batteries. Therefore, these devices do not support traditional security mechanisms, so they remain very vulnerable so far. In addition to the difficult implementation of traditional security mechanisms, other factors contributing to this vulnerability include: **1-**The use of wireless communications for mobility needs. **2-**Manufacturers’ interactions with devices, e.g., firmware updates and information gathering and sharing, which expose the latter to attacks. **3-**Flimsy design in terms of security due to the limited expertise and experience of young companies that often focus more on the services provided by these devices than on their security.

Despite this high level of vulnerability, the number of IoT devices continues to grow and has already reached several billion devices worldwide. Sometimes deployed within critical network infrastructure in industrial and residential environments, these devices make the networks they are a part of more vulnerable than ever because they are prime targets for attackers who use them as gateways to reach more important devices and networks. An illustration of this reality is the Mirai botnet [1] which was an unprecedented-scale attack on IoT devices in September 2016. Mirai botnet was designed to perform distributed denial-of-service (DDoS) attacks on servers around the world with a previously created army of IoT device bots (an IoT botnet). Several papers in the literature [2,3,4,5,6,7] show an overview of the incredible number of possible attacks on IoT devices. 

Because of their individual vulnerability and the risk to the entire network infrastructure they are a part of, there is a need to find new security mechanisms adapted to these devices, and this is something researchers have been working on for some years now. However, the industry of IoT devices is, so far, poorly governed by standards. Consequently, a network of these devices can be very heterogeneous in terms of communication protocols, including possible non-open-source protocols. This heterogeneity implies that the most suitable security solutions for IoT device networks must be generic solutions, i.e., protocol-independent solutions. Thus, one of the interesting state-of-the-art approaches, which is the scope of this paper, is fingerprinting. In [8], Xu et al. present an overview of fingerprinting and its challenges and opportunities.

To prevent a malicious device from impersonating a legitimate one in order to perform man in the middle (MitM) or spoofing attacks, fingerprinting is based on the assumption that two perfectly identical devices do not exist because of errors introduced by the manufacturing processes. The purpose of the fingerprinting is to take advantage of these small hardware imperfections to identify the devices. These hardware imperfections can affect, for example, the real (practical) speed of clocks, the accuracies of digital-to-analog/analog-to-digital converters, the precision of IQ constellation values (IQ imbalances), etc. The slight difference between the theoretical and practical values of these parameters can be used as fingerprinting features. The efficiency of the fingerprinting approach relies on the relevance of these features used to build the device fingerprint. One of the most important relevance criteria of the fingerprinting features is their resilience to the surrounding physical phenomena over time. If these features change over time and with environmental conditions, the fingerprint built from them will also change over time and with ambient conditions. Therefore, such a fingerprint cannot be used to identify a device and the features used to build the fingerprint are considered not relevant.

In this work, we answer the question of whether or not the PSD of the RF signal transmitted by a device could be a relevant feature for building the fingerprint of this device for identification purposes. By carrying out various experiments, we were able to observe the behavior of PSD considering different environmental conditions and different data transmitted by the device. We then highlighted the limitations and possibilities of PSD as a fingerprinting feature, which is the major contribution of this paper. It intends as a support to those who will attempt to use the PSD, for example, as a feature for fingerprinting and intrusion detection systems (IDSs). Features are criteria that fingerprinting uses to establish the identity of a transmitter or that IDS uses to establish the legitimate nature of a communication.

In addition, this paper provides a precise overview of fingerprinting by highlighting the different approaches of the literature and their limitations; the paper also outlines the opportunities and main challenges when using fingerprinting as a cybersecurity solution.

## 2. Related Work

Searching for new security mechanisms for IoT devices, researchers have been working on the fingerprinting approach for some years now. Several papers about this approach have already appeared in the literature, highlighting its opportunities and challenges.

In [9], Chatterjee et al. present “RF-PUF” for radio frequency physical unclonable functions. This expression “Physical Unclonable Functions”, introduced by Gassend et al. [10,11], basically refers to any deviation from the theoretical data on the electronic hardware due to the manufacturing processes. These imperfections, whose impacts are not a concern in the normal use of the hardware, are assumed to be unique and inherent to each electronic piece, even for two instances of the same product manufactured on the assembly lines at the same time. These inherent imperfections would, hence, be for electronic hardware what DNA, fingerprints, or voices are for human beings, which is why they would be useful for electronic hardware identification. Fingerprinting features are nothing but those deviations from the theoretical values. For a wireless communicating circuit, these deviations can be extracted at two different levels and be used for the fingerprinting of this circuit. **1-**We can focus on the deviations in some specific components of the circuit such as clocks, ADC/DAC, amplifiers, and so on in order to perform in-situ fingerprinting on the concerned device. **2-**We can focus on the deviations in the RF signal emitted by the circuit and perform centralized fingerprinting with the help of an RF receiver. Remember that the deviations in the RF signal emitted by the circuit are in fact the echo of the deviations in the components of the circuit.

Like Chatterjee et al. [9], we chose the second option in this work. However, while we were interested in the deviations in the PSD, Chatterjee et al. [9] chose the frequency offset, the I-Q imbalances, and channel features for compensation (attenuation, distortion, and Doppler shift) as fingerprinting features.

Despite a certain efficiency of their solution, Chatterjee et al. [9] illustrated in an experiment one of the major challenges that fingerprinting faces, which is the instability of the fingerprinting features with the environmental parameters. Indeed, changing the temperature of their experimental closed environment in discrete steps of 5 °C in the range of 0 °C–25 °C modified the transmitter (emitted RF signal) properties, and the receiver (“the fingerprinter”) considered every 5 °C change in the temperature as a different transmitter. This means that the used fingerprinting features were not resilient to environmental parameters, which is the temperature in this case. This is a major limitation of their solution, which is finally unusable as it is in a real-world IoT environment, as the same device fingerprint can change from one moment to another depending on the environmental conditions.

In [12], Wang et al. presented RF-DNA (radio frequency–distinct native attribute) which can be considered as another name for RF-PUFs. They chose amplitude, phase, and instantaneous frequency as fingerprinting features. 

In their experiments, the accuracy of the devices’ fingerprinting identification dropped drastically when the noise level increased in the experimental environment.

In [13], Vaidya et al. performed in-situ fingerprinting on commercial-off-the-shelf (COTS) components including Arduino boards. They use clock oscillator and analog-to-digital converter (ADC) deviations as fingerprinting features.

However, in-situ fingerprinting schemes are generally difficult to deploy in a real-world IoT environment because all the devices in the network have to run a local copy of the fingerprinting software, which is a major limitation of these schemes since a real-world IoT environment is made of devices that are generally not designed to be rewritable by users. Moreover, an IoT environment can be very heterogeneous in terms of communication protocols and very dynamic with BYOD (bring your own device) devices coming in and out.

In the literature, authors propose other fingerprinting schemes including the addition of external circuitry dedicated to device-specific fingerprint generation, such as the addition of an extra layer of silicon by chip manufacturers or the addition of an external circuit by a specialist. This solution not only adds extra costs but also requires the modification of the hardware architecture of an out-of-box IoT device purchased on the market, which is not convenient.

There are still other ideas in the literature about fingerprinting. Electronic devices emit transient signals during the boot-up phase. Some authors are interested in using these transient signals as fingerprinting features [14,15]. This method is again less suitable for a real-world IoT environment where most of the devices added to it are already fully powered on.

Concerning PSD, in which we are interested in this work, to the best of our knowledge, Galtier et al. presented in [16] the first works on the use of PSD in security with the fingerprinting approach. However, with the attacker model and deployment environment considered, the authors did not take into account the possibility that the communication devices had very similar characteristics, nor that the environment could change a lot. Their solution was designed for static environments where the devices do not move and the ambient conditions remain fairly constant, which is not the case for most real-world IoT environments. In this survey, we will highlight the limitations of PSD in a dynamic environment.

## 3. Context and Threat Model

Let us consider a critical network infrastructure made of various IT systems including IoT devices. Let us assume that all network devices are physically reliable, i.e., their structure cannot be modified. The IoT devices wirelessly exchange confidential data, e.g., health data from a smartwatch, or critical data, e.g., a time set point for a chemical process or a temperature set point for a communicating thermostat. These data need to have integrity in order to counter potential malicious actions through data corruption that could have severe consequences. Thus, to guarantee this integrity, a solution could be the identification of the transmitter of the data in order to know whether or not it is a legitimate transmitter. Indeed, without such a solution and regarding the very nature of the wireless transmission medium, an attacker can transmit corrupted data or unwanted instructions to a device, for instance, the wireless transmission of an unwanted temperature set point to a communicating thermostat in order to increase the temperature to an unreasonable level.

With physical access to the network infrastructure and equipped with a malicious device that they fully control, with no need to even integrate that device into the network, the attacker can communicate with the IoT devices in the network through the wireless medium, which opens the way for malicious actions such as transmitting unwanted instructions and corrupted data. Being in such a position, the attacker can perform a plethora of attack scenarios against these devices that can lead to a breach among the three major security aspects, that are: **1-**making a device unable to perform its purpose through attacks such as denial of service (DoS), power off, and so on, which are breaches of its availability; **2-**gathering sensitive information, which is a breach of confidentiality; and finally, **3-**modifying the device, reducing its capabilities, or diverting its usage [17], which are breaches of its physical and/or software integrity.

To counteract some of these attacks that take advantage of the propagating nature of radio waves in order to establish countermeasures and to ensure the integrity of data exchanged between devices, we must be able to identify the transmitter of the data even if these data are meaningful for the targeted device. We aim to carry this out with the RF fingerprinting approach using the PSD as a fingerprinting feature.

## 4. Analytic Development

### 4.1. PSD and Correlation Function

Consider a transmitted signal $x\left(t\right)$, proposed for simplicity, a periodic power signal. Then it can be written as
(1)$$x\left(t\right)=\overset{\infty }{\underset{n=-\infty }{\sum }}C_{n}e^{i2\pi f_{n}t}\;$$
where $f_{n}=nf_{0}$ and $f_{0}=1/T$ with T the period of x. The autocorrelation function $R_{x}\left(\tau \right)$ is defined as
(2)$$\frac{1}{T}\int_{\left(T\right)}^{\;}x\left(t\right)\;\bar{x\left(t+\tau \right)}\;dt$$
where $\bar{x}\;$denotes the complex conjugate of x. A simple calculation leads to
(3)$$R_{x}\left(\tau \right)=\overset{\infty }{\underset{n=-\infty }{\sum }}{\left|C_{n}\right|}^{2}\;e^{i2\pi f_{n}\tau }\;$$

The power density function of x is the Fourier transform of $R_{x}\left(\tau \right)$ and is given by
(4)$$PSD_{x}\left(f\right)=TF\left[R_{x}\left(\tau \right)\right]=\overset{\infty }{\underset{n=-\infty }{\sum }}{\left|C_{n}\right|}^{2}\delta \left(f-f_{n}\right)$$

### 4.2. PSD through a Channel

Consider a multipath fading channel of N paths. The channel output which is the input signal for the receiver, can be written as in [18]
(5)$$y\left(t\right)=\overset{N}{\underset{k=1}{\sum }}a_{k}\left(t\right)\;x\left(t-\tau_{k}\left(t\right)\right)$$
where $a_{k}\left(t\right)$ and $\tau_{k}\left(t\right)$ are, respectively, the attenuation and the propagation delay associated with the *n*th multipath component. If the transmitter and receiver are at rest and if there is no change in the transmission medium, the transmission delay associated with the *n*th propagation path and the path attenuation are constant, that is $\tau_{k}\left(t\right)=\tau_{k}$ and $a_{k}\left(t\right)=a_{k}$. In that case, we have
(6)$$y\left(t\right)=\overset{N}{\underset{k=1}{\sum }}a_{k}\;x\left(t-\tau_{k}\right)$$

The channel is fixed and can be represented by its impulse, i.e.,
(7)$$h\left(t\right)=\overset{N}{\underset{k=1}{\sum }}a_{k}\delta \left(t-\tau_{k}\right)\;$$
or its transfer function
(8)$$H\left(f\right)=\overset{N}{\underset{k=1}{\sum }}a_{k}e^{-i2\pi f\tau_{k}}\;$$

The output signal $y\left(t\right)$ is then obtained through the following convolution product
(9)$$y\left(t\right)=\int_{-\infty }^{\infty }h\left(u\right)\;x\left(t-u\right)\;du\;$$
and the PSD being the Fourier transform of the autocorrelation of $y\left(t\right)$, we have
(10)$$PSD_{y}\left(f\right)={\left|H\left(f\right)\right|}^{2}PSD_{x}\left(f\right)=\overset{\infty }{\underset{n=-\infty }{\sum }}{\left|C_{n}\right|}^{2}{\left|H\left(f_{n}\right)\right|}^{2}\delta \left(f-f_{n}\right)\;$$

But
(11)$$\begin{array}{l} {\left|H\left(f\right)\right|}^{2}=H\left(f\right)\bar{H\left(f\right)}=\overset{N}{\underset{k=1}{\sum }}a_{k}e^{-i2\pi f\tau_{k}}\;\overset{N}{\underset{l=1}{\sum }}a_{l}e^{i2\pi f\tau_{l}}\\ =\overset{N}{\underset{k=1}{\sum }}a_{k}{}^{2}+\overset{N}{\underset{k=1}{\sum }}\;\overset{N}{\underset{l=k+1}{\sum }}a_{k}a_{l}\left(e^{-i2\pi f\left(\tau_{k}-\tau_{l}\right)}+e^{i2\pi f\left(\tau_{k}-\tau_{l}\right)}\right)\\ =\overset{N}{\underset{k=1}{\sum }}a_{k}{}^{2}+2\overset{N}{\underset{k=1}{\sum }}\;\overset{N}{\underset{l=k+1}{\sum }}a_{k}a_{l}\cos\left(2\pi f\left(\tau_{k}-\tau_{l}\right)\right)\\ =\left[\begin{array}{ccc} a_{1} & \cdots & a_{N} \end{array}\right]\underset{M\left(f\right)}{\underset{︸}{\left[\begin{array}{cccc} 1 & \cos\left(2\pi f\left(\tau_{1}-\tau_{2}\right)\right) & \cdots & \cos\left(2\pi f\left(\tau_{1}-\tau_{N}\right)\right)\\ \cos\left(2\pi f\left(\tau_{1}-\tau_{2}\right)\right) & 1 & \cdots & \cos\left(2\pi f\left(\tau_{1}-\tau_{N}\right)\right)\\ \vdots & \vdots & \vdots & \vdots \\ \cos\left(2\pi f\left(\tau_{1}-\tau_{N}\right)\right) & \cos\left(2\pi f\left(\tau_{2}-\tau_{N}\right)\right) & \cdots & 1 \end{array}\right]}}\;\left[\begin{array}{c} a_{1}\\ \vdots \\ a_{N} \end{array}\right]\; \end{array}$$

Note that because ${\left|H\left(f\right)\right|}^{2}\geq 0$, the symmetric matrix $M\left(f\right)$ is positive semidefinite for all f.

Analyzing the off-diagonal terms of Hfn2, their absolute values cos2πnf0τk−τl are sensitive to the values of the delays τk and are close to one if
(12)τk−τl≈j2f0=jT2, j∈ℤ 

As expected, the PSD of signal$\;y\left(t\right)$ can be significantly affected by the channel and its use as a fingerprinting feature is really questionable, even more so when the channel characteristics are changing due to the changes in the transmission medium.

## 5. Methodology

### 5.1. Aims of This Work

An ideal fingerprinting solution would be one that is able to build a unique and stable fingerprint for a device from its inherent imperfections introduced by the manufacturing processes. This fingerprint should remain the same for the entire lifetime of the device, it should not change in any circumstances. In such a case, we consider as relevant the fingerprinting features (i.e., inherent imperfections) used to build this fingerprint.

Since these inherent imperfections reflect on the RF signal that the device emits, we aim to figure out whether or not the PSD of this RF signal would be among the relevant fingerprinting features, meaning unique for a given device and stable under all circumstances. We will, hence, examine the behavior of the PSD regarding two influential elements, namely:Ambient conditions (temperature, relative humidity, etc.);The data carried by the emitted RF signal, i.e., the data modulating the RF carrier.

These elements are two among others (e.g., device aging) but are enough to address our question of whether or not the PSD would be a relevant fingerprinting feature for device identification.

### 5.2. Experiments Overview

To assess the influences of the environment and data on the PSD of the RF signal, we carried out a certain number of experiments. These experiments were performed on Bluetooth low energy (BLE) devices designed at LAAS-CNRS [19,20] as part of a project that aims to monitor the structural health of reinforced concrete throughout its lifetime. To ensure that, these devices measure some physical parameters (e.g., temperature, relative humidity, and resistivity) of reinforced concrete and then wirelessly transmit these data via the Bluetooth low energy (BLE) technology for processing by a data collector.

In this paper, we are interested in the PSD of the BLE signal emitted by these devices. Their hardware and software architectures have evolved over time and, today, we have three different versions.

To address the influence of environmental conditions on the PSD, we performed experiments in two different environments, namely: **1-**An anechoic chamber which is an external-wave-isolated chamber; the electronic noise is, therefore, very low in this chamber. **2-**In open space, where the environmental parameters are less favorable to the propagation of the electromagnetic waves.

To address the influence of the data modulating BLE RF carrier on the PSD, we performed experiments with different data by performing a spoofing attack on the BLE device, since the identifier is part of the transmitted data. We configured all the BLE devices to the same identifier in order to have them transmit exactly the same data, and we compared the PSDs. We also looked at the case of a similar BLE device when it is experimented on with two different identifiers.

The Table 1 below summarizes the experiments in our study. “Experiment 1” aimed to evaluate the influence of environmental parameters on the PSD. For this purpose, we compared the PSDs of an experiment when carried out in the anechoic chamber with the PSDs of the same experiment when carried out in open space. In this experiment, all the devices kept their original identifier, contrary to the other two experiments.

Then, “Experiment 2” and “Experiment 3” aimed to evaluate the influence of the data on the PSD; they involved data modulating the RF carrier of the BLE, i.e., the transmitted data.

For this purpose, we changed the identifier on a similar BLE device in “Experiment 2” and we assigned a similar identifier to four BLE devices in “Experiment 3” knowing that the identifier was part of the data transmitted by the device.

In “Experiment 3”, the devC2 device was replaced by the dev09 device because the transmission power of devC2 (the first ever version) was very low compared to the other devices.

This difference would make our PSD comparison more difficult and less relevant because, as will be demonstrated, the transmission power has an effect on the PSD.

### 5.3. PSDs Comparison

To compare PSDs, we needed a method that can give the difference between them in order to know how close or different they are. 

Let us consider two signals, $s_{1}$ and $s_{2}$, whose PSD spectrums are $PSD_{1}$ and$\;PSD_{2}$, respectively. The spectrums $PSD_{1}$ and $PSD_{2}$ are in the same frequency band named *B*.

To compare these two PSDs, we chose a method that is outlined in Figure 1 below.

Within band B (Figure 1), the below conditions (15), (16), (17), and (18) are checked at N frequencies spaced from one to the next by Δf. The parameter$\;d=\left|PSD_{1}^{*}\left(f\right)-PSD_{2}^{*}\left(f\right)\right|$ is compared to ε in condition (15). For band *B*, we chose 2 MHz, which is the channel bandwidth of Bluetooth low energy (BLE).

This comparison method can be presented in two steps as follows.

#### 5.3.1. PSD Normalization

As will be demonstrated, the amplitude of the PSD varies with several parameters (e.g., the distance between the transmitter and the receiver); hence, a comparison approach more focused on amplitude will be less relevant for comparing two PSDs for fingerprinting purposes, especially in a dynamic IoT environment. For this reason, with the following Formula (13), we will first normalize each of our two PSDs to their respective maximum amplitudes.
(13)$$\forall f\in B,\;PSD^{*}\left(f\right)=PSD\left(f\right)\;/\;|PSD_{max}|$$
where$\;PSD\left(f\right)$ denotes the amplitude of the spectrum at frequency f, $PSD_{max}$ denotes the maximum amplitude of the spectrum, and $PSD^{*}\left(f\right)$ denotes the normalized amplitude at frequency f.

#### 5.3.2. Comparison

So now the two PSDs have been normalized, their amplitude difference is no longer a concern. We are going to use a comparison method which is based on the following idea: provided that they are in the same frequency band, that is B, without any frequency offset between them, the two spectrums$\;PSD_{1}$ and $PSD_{2}$ have the same profile if and only if:(14)$$\forall f\in B,\;PSD_{1}^{*}\left(f\right)=PSD_{2}^{*}\left(f\right)\;$$
where$\;PSD_{1}^{*}$ and$\;PSD_{2}^{*}$ denote the normalized amplitudes of the PSD spectrums of signals $s_{1}$ and $s_{2}$, respectively.

However, Equation $\left(14\right)$ represents an ideal scenario where the two PSDs are perfectly identical, so it is a rather theoretical equation and overoptimistic. In practice, we will allow an acceptable tolerance ε as follows:(15)$$\forall f\in B,\;d=\left|PSD_{1}^{*}\left(f\right)-PSD_{2}^{*}\left(f\right)\right|\leq \;\varepsilon$$

Note that the comparison does not include all frequencies f∈ℝ in band B but rather addresses some specific frequencies defined by the PSD estimation function, i.e., the Welch periodogram method. Nevertheless, these considered frequencies increase from one to the next by a fixed step that we call Δf. This Δf is the frequency resolution of the comparison; the lower Δf, the more accurate the comparison because the number of considered frequencies is higher. This frequency resolution Δf is defined as well by the PSD estimation function.

However, at a given frequency f, the fulfillment of condition (15) is necessary but not sufficient to claim that the two PSDs have the same profile between f and f+Δf. Indeed, condition (15) could be satisfied in f while the two PSDs evolve in different directions between f and f+Δf, e.g., $PSD_{1}$ rises while$\;PSD_{2}$ falls or vice versa, which is a strong proof of the two PSDs being different. To take into account this important possibility, one of the three conditions below must be satisfied in addition to condition (15).
(16)$$PSD_{1}^{*}\left(f\right)>PSD_{1}^{*}\left(f+\Delta f\right)\;\&\&\;PSD_{2}^{*}\left(f\right)>PSD_{2}^{*}\left(f+\Delta f\right)$$
(17)$$PSD_{1}^{*}\left(f\right)<PSD_{1}^{*}\left(f+\Delta f\right)\;\&\&\;PSD_{2}^{*}\left(f\right)<PSD_{2}^{*}\left(f+\Delta f\right)$$
(18)$$PSD_{1}^{*}\left(f\right)=PSD_{1}^{*}\left(f+\Delta f\right)\;\&\&\;PSD_{2}^{*}\left(f\right)=PSD_{2}^{*}\left(f+\Delta f\right)$$
where && denotes the logical “AND” condition.

These three conditions are actually a simplified form of the comparison of the first derivatives between f and f+Δf of the two PSDs. These conditions are sufficient in this survey since the frequency resolution of the comparison is quite good.

Hence, we consider that the two PSDs have the same profile between f and f+Δf when condition (15) is satisfied and one of the conditions (16), (17), or (18) is also satisfied, i.e., 15 AND (16 OR 17 OR 18). Thus, after going through the entire band B, we compute a similarity percentage $S_{p}\;$between the two PSDs with the following formula:(19)$$S_{p}=\frac{N_{F}}{\frac{B}{\Delta f}+1}\times 100\;\;S_{p}=\frac{N_{F}\times \Delta f}{B+\Delta f}\times 100\;$$
where $N_{F}$ is the number of frequencies, out of a total of N, for which condition (15) and one of the conditions (16), (17), or (18) are satisfied.

The higher $S_{p}\;$, the closer the profiles of the two PSDs, provided that they are both in the band B, there is no frequency offset between them, and, finally, the value of ε in condition (15) is acceptable. Indeed, $S_{p}$ depends on the value of ε, whose choice is empirical. The lower ε, the more relevant the comparison result, whereas the higher ε, the higher the similarity percentage could be, but it will not represent the reality. For example, if we ignore conditions (16), (17), and (18) and assume ε=∞, the similarity percentage$\;S_{p}$ will still be equal to 100% but will have no meaning.

This approach allows us to compare the profiles of the PSDs regardless of the difference between their amplitudes, which is more relevant in a dynamic IoT environment where the devices move a lot, knowing that the amplitude of the PSD is dependent on the distance between the transmitter and the receiver. However, a disadvantage of this approach could be the high number of computations to perform. Indeed, the greater the number N of considered frequencies, that is, the smaller Δf, the higher the number of computations to be performed, although the comparison is more accurate.

## 6. Experiments and Results

### 6.1. Experimental Setup

The key elements of our experiments are BLE devices, shown in Figure 2 below. The aim of this work was to find out whether it would be possible or not to identify the BLE device that emitted a signal thanks to the PSD of said received BLE signal, and whether this identification remains possible in any environmental conditions and independently of the data modulating the BLE carrier.

In our experiments, we used the RSA306B real-time spectrum analyzer with SignalVu-PC software for receiving and recording the BLE signal as IQ samples. Then, we used these data in a python implementation of Welch’s average periodogram method [21] to estimate the power spectral density. Throughout our experiments, we kept the distance between the transmitter (i.e., the BLE device) and the receiver (i.e., RSA306B) at 2 m for two reasons, namely: **1-**the dimensions of our anechoic chamber made it difficult to exceed this distance; **2-**fortunately, regarding what we were trying to prove about the use of a PSD as a fingerprinting feature, it was not necessary to experiment with different distances between the transmitter and the receiver in this survey. Indeed, based on the results of the 2 m experiments, we could already explain the behavior of the PSD with respect to the distance between the transmitter and the receiver. More details on this assertion are given in “Section 7”.

Figure 3 shows the experimental setup in the anechoic chamber, which was the same setup used in open-space experiments, and Figure 4 schematizes the experimentation in a more readable way. 

In Figure 3, we can observe that the experimentation was quite simple and did not require a lot of equipment. It was, therefore, an easily reproducible experiment and, for this reason, we are going to give some configuration information about certain components in order to guide those who will try to reproduce this experiment.

We configured the BLE devices in nonconnectable advertising mode [22], where they advertise information to any listening device.

It is important to know that any transmission of information by a BLE terminal (i.e., via the BLE protocol) starts with a step called the advertising phase, during which the terminal broadcasts information that contains its intentions such as a request to establish a connection with another terminal or a message broadcasted to any nearby listening terminal, as was the case for our BLE devices in this experiment.

Among the 40 channels used by the BLE in the 2.402 GHz to 2.480 GHz band, 3 channels are exclusively dedicated to advertising and are called primary advertising channels. At every advertising event, the advertising information is transmitted on these 3 channels one after another.

We, therefore, configured our receiver listening to one of these channels to capture and record the BLE signal each time we triggered the advertising on the BLE device.

These devices were similar to BLE beacons where we controlled the triggering of the advertising as well as the advertising data. Thanks to this, we were able to measure the change in the data modulating the BLE carrier.

The advertising mode was more suited to our experimentation because it allowed us to avoid the frequency-hopping mechanism used by the BLE protocol when the communication involved the 37 other channels. Indeed, with the frequency selection pattern used by the hopping mechanism being random, the PSD of the signal of the same message changed with the change in the frequency selection pattern, which could have misled us in our analysis of the PSD. More information about this BLE protocol, widely used in IoT, can be found in [23,24].

The second component to consider in this experiment was the RSA306B real-time spectrum analyzer. This device allowed us to capture and record the BLE signal as IQ samples in a CSV format file. This device allowed to listen in real-time mode up to 40 MHz bandwidth, which was more than enough for our application. Beyond this bandwidth, it works in sweep mode. Some of the most important configuration parameters of this device are: the center frequency to be listening, the span which must be higher than the BLE channel bandwidth (i.e., higher than 2 MHz), the acquisition triggering setup, the acquisition duration, and the sampling frequency. For our experiment, the values of these parameters are given in the Table 2 below.

To guarantee the integrity of our results, we had to record the entire advertising signal. To do so, we had to choose a convenient acquisition duration, that is, longer than the duration of the advertising on a channel. To find the duration of the advertising on a channel, we observed the power consumption of the BLE device during advertising.

Figure 5 below shows the profile of the power consumption by the BLE device during an advertising period; the three peaks represent the three channels. We can see that each peak corresponding to the advertising on a channel lasted only about 0.5 ms. This is why we chose to perform the acquisition over 1 ms from the occurrence of the trigger event; hence, 1 ms is the value of the parameter “Acquisition Length”. This was more than enough time to acquire the entire advertising signal on a channel, as witnessed by all the PSD plots in this paper.

It is worth mentioning that the order of advertising on the three channels was random, although this did not affect our acquisition which was only triggered when the advertising arrived on the channel we were listening to, i.e., the second channel.

### 6.2. Results

In this section, we will present the results of our experiments.

First of all, we will present an example of 20 PSDs (20 transmissions) from one of the BLE devices at 2 m from the RSA306B receiver in order to check for possible dynamics on the PSD from one transmission to another while any parameters (environment, data, distance, etc.) have not been changed. 

This section will also present the practical application of our comparison approach.

Next, we will move on to the results of our experiments starting with “Experiment 1”, the comparison of the PSDs from anechoic chamber experiments with those from open-space experiments.

We will close this section with the results of the experiments “Experiment 2” and “Experiment 3” by examining the PSDs for different data, that is, data modulating the BLE carrier, i.e., the data transmitted by the BLE device.

#### 6.2.1. Experiment 0: Example of 20 PSDs’ Acquisition

Figure 6 below shows 20 PSDs from the same device in a static experimental setting, i.e., no parameter changed during the 20 transmissions.

We can notice that the plot formed by the 20 PSDs was quite thin especially in the band of interest, *B*. We chose signals captured in an anechoic chamber because if their PSDs showed differences, then this dynamic would be worse in open space where there is more noise than in an anechoic chamber.

At first glance, one could conclude that these 20 PSDs had the same profile. But to be sure, we are going to compare each one with another one, that is to say 10 pairs in total. Table 3 below shows the results of this comparison knowing that the region between the two red lines (Figure 6) is the one considered by the comparison script.

The width of this region, which we call the band *B*, is: *B*(Hz) = 2,427,011,718.75 − 2,424,988,281.25 = 2,023,437.5. The frequency resolution Δ*f* computed by the PSD estimation algorithm is Δ*f*(Hz) = 27,343.75. Thus, our comparison algorithm checks whether condition (15) and one of conditions (16), (17), or (18) are satisfied or not at *N* = (*B*/Δ*f*) + 1 = 75 frequencies within interval [2,424,988,281.25, 2,427,011,718.75] (Hz) for two values of *ε*, namely: *ε*(dB) = 0.025 and *ε*(dB) = 0.05. Remember that these values are completely empirical. The smaller *ε*, the more severe the comparison.

The parameter $N_{F}$ is the number of frequencies, out of a total of *N* = 75, at which condition (15) and one of conditions (16), (17), or (18) are satisfied.

Ideally, the percentage of similarity $S_{p}$ should have the value of 100% in all the cases of Table 3 but we can see that there is no case of a 100% similarity percentage according to our algorithm. This is the proof that even in a static experimental setting, the PSD may have some dynamics, i.e., the PSD may have minor deviations.

However, one can notice that the value of ε had only a slight influence on the value of$\;S_{p}$ in this case, which makes sense since the 20 transmissions were being performed at the same distance from the receiver and in the same environmental conditions, so the amplitude variation was quite small. In this case, the changes in the value of *S_p_* came mainly from the minor variations in the profile of the PSDs.

Note that the values of band B, frequency resolution Δf, and, therefore, N=75 remained the same for all PSD comparisons in this paper.

#### 6.2.2. Experiment 1: PSD Measurement in Various Environment

The two graphs (a) and (b) on Figure 7 below show the PSDs of the BLE signal for different BLE devices.

In each graph, each color actually represents a set of 20 plots (i.e., 20 PSDs) corresponding to 20 transmissions of the same data by the same BLE device at the same position and distance from the receiver (RSA306B). This is true for each color in a graph in this paper.

From one graph to the other, plots of the same color represent the same BLE device, e.g., the green plots in graph (a) and the green plots in graph (b) are all PSDs of the device dev48, while plots of graph (a) correspond to the transmission in the anechoic chamber and the ones of graph (b) correspond to the transmission in open space.

It is also worth remembering that, in this experiment, the transmitted data are different from one device to another since the BLE device identifier is part of these data. All devices have their original unique identifier without any change. Graph (a) shows the results of the experiment in the anechoic chamber and graph (b) shows the results of the same experiment in open space. 

We can notice that the amplitudes of the plots on graph (a) are higher than the amplitudes of the plots on graph (b); for example, the green plots have an amplitude of around −69.5 dB on graph (a) and −75.5 dB on graph (b). In fact, the amplitude of the PSD reflects the amount of power received by the receiver; the higher the received power the higher the amplitude of the PSD. The lower the received power, the lower the amplitude of the PSD, and the more the PSD’s profile changes.

This difference in amplitudes between the graph (a) plots and graph (b) plots means that under the same conditions except environmental, the amount of power arriving at the receiver in the anechoic chamber is higher than the amount of power arriving at the receiver in open space. This is already noticeable at only 2 m between the transmitter and the receiver.

It can also be noticed that in Figure 7, in the case of the same device (plots of the same color from one graph to the other), the graph (a) PSD plots are thinner than the graph (b) PSD plots. This means that the amount of power arriving at the receiving antenna from one transmission to another is more stable in the anechoic chamber than in open space. The channel has a significant influence on the signal and does not allow for stable PSDs.

Despite this evidence, we made comparisons using our comparison algorithm. The goal of this experiment was to analyze the influence of environmental parameters on the PSD of a given device; hence, comparing the PSDs of two different devices was of no interest. Instead, what was interesting was to compare the PSDs from the anechoic chamber with ones from open space for the same device and this is what we carried out. Hence, we aimed to compare the plots of the same color from one graph to the other. As previously mentioned, each color on a graph is actually a set of 20 plots, so we compared the averages rather than performing a 20 versus 20 comparison. Therefore, the process consisted of calculating the average of one of the groups of 20 PSDs of a graph, normalizing this average and, finally, comparing it with the normalized average of the same group of 20 PSDs of the other graph, with the compared PSDs being the same color on the two graphs. For instance, for dev48, we compared the normalized average of the green plots in graph (a) with the normalized average of the green plots in graph (b). The table below (Table 4) shows the results of this comparison.

These results (Table 4) show that the higher the difference of amplitudes, the more influential ε is and the more the profiles of the PSDs are different according to our comparison algorithm. Indeed, in Figure 7b, one can notice that the lower the amplitude, the more abrupt the slopes of the plots, and the greater the change in the profile of the PSD. This fact is related to the hardware and computing methods of the PSD and not related to the BLE device or signal. When the amplitude decreases, a sort of compression occurs which causes the details of the plot to disappear, i.e., the PSD’s profile is changing.

These results highlight the effects of environmental parameters on the PSD of the received signal. Increasing the distance between the transmitter and the receiver would amplify these effects of environmental parameters on the PSD even more. Indeed, in the case of RF wireless communications, any parameter influencing the amount of power arriving at the receiving antenna also influences the PSD. The transmitted power, the distance between the transmitter and receiver antennas, the positioning of the transmitting antenna in relation to the receiving antenna regarding polarizations, multipath propagation and fading effects, the temperature, the relative humidity, and interferences with other transmitters are some of these parameters.

However, observing graphs (a) and (b), one can notice that the profile of the PSD is not as sensitive to the change in environmental parameters as its amplitude is.

#### 6.2.3. Experiment 2: PSD vs. Transmitted Data, a Case of the Same Device Transmitting Two Different Identifiers

The physical parameters, including temperature, measured by our BLE devices were not stable and could change on the fly according to the weather. In order to always transmit exactly the same data over time, we assigned constant values to these parameters in the firmware and, thus, ignored the values measured by the sensors. In this way, we performed data change experiments, that is, this experiment (Experient 2) and the next one (Experiment 3), by changing the identifier of the BLE device as the identifier was part of the data transmitted by the device.

In this experiment (Experiment 2) performed inside of an anechoic chamber, we used one and only one BLE device. Firstly, we performed the experiment using the device with its original identifier without any change. Secondly, we changed only the device identifier and we repeated the same experiment. The results are shown in Figure 8 below.

The two groups of 20 plots each on Figure 8 are from the same BLE device. The blue plots are PSDs of the device when using its original identifier (E0512496C3CA) and the dark-orange plots those of the same device when using the device dev48’s identifier (E05124969648). We have spoofed the identity of the device dev48.

We compared the PSDs (the average of the 20 PSDs) of the device when it had its real identifier with those when it had the identifier of another device. In reality, we compared the normalized average of the blue plots with the normalized average of the dark-orange plots. Table 5 below shows the results of this comparison.

In this comparison, ε, again, had less influence on$\;S_{p}$. This makes sense as we can see in Figure 8 that the PSDs had almost the same amplitude, so the value of$\;S_{p}$ was more impacted by the difference between the profiles than the difference in amplitudes. So, although being the same device, the PSDs of the device when using its original identifier were different from the PSDs of the same device when using another identifier. The part circled in red on Figure 8 highlights some apparent differences. Remember that the identifier is part of the transmitted data.

One can already notice that this time, the profile of the PSD was more sensitive to the changes in data compared to its amplitude. Meanwhile, we were able to witness in the previous experiment (Experiment 1) that the amplitude of the PSD was more sensitive to environmental conditions compared to its profile.

The next experiment (Experiment 3) was a generalized form of this experiment (Experiment 2). In “Experiment 3”, we assigned the same identifier to four different BLE devices, in order to make them always transmit exactly the same data.

#### 6.2.4. Experiment 3: PSD vs. Transmitted Data, a Case of Four Different Devices Transmitting with the Same Identifier

In this experiment, we carried out identity spoofing by giving the identifier of one of the 4 BLE devices to the 3 others. We ended up with 4 BLE devices always transmitting exactly the same data. The results are shown in Figure 9 below.

The plots on the graph in Figure 9a are an overlapping of the 4 groups of plots on the graph in Figure 9b. In Figure 7a, where the transmitted data are different from one device to another, we can see that the PSDs are much more distinguishable than they are in Figure 9a, where devices are transmitting the same data. As usual, each color’s plot is actually a set of 20 plots corresponding to 20 PSDs, i.e., 20 transmissions, from the same device.

Here, we compared the PSDs, that is, again, the normalized average of the 20 PSDs of each device with those of the three others, that is, a combination of 2 among 4; hence, 6 pairs of PSDs, as shown in Table 6 below:

These results in Table 6 are very interesting. One can notice that the comparisons in which devCA is involved have low similarity percentages more often and these cases are also where the value of ε is the more influential; this is due to the frequency offset of devCA regarding the other devices. For instance, Figure 10 below shows the PSDs of dev48 (green plots) and devCA (brown plots), and we can see that the PSDs of devCA are shifted from dev48’s PSDs.

We could have corrected this frequency offset before comparing the PSDs but we chose to show it because it may be an interesting point for the readers of this paper. What is interesting about the frequency offset is that it is inherent to the devices; it has even been used as a fingerprinting feature in some surveys, for instance, in [9]. 

The other parameter that we could consider is amplitude, but this had almost no influence on the results of this comparison since the 4 average PSDs had almost the same amplitude. Indeed, the average PSD amplitudes were −72.23 dB, −72.88 dB, −72.07 dB, and −72.90 dB for dev48, devCA, dev02, and dev09, respectively.

So, the main factor that made the results in Table 6 not an exact representation of the reality was the frequency offset. However, the profiles of the PSDs were not identical either; it can be seen, for example, that there is a hollow at the center frequency (2.426 GHz) of the green plot. These slight differences may actually come from hardware imperfections of the devices (local oscillators, filters, power amplifiers, etc.). However, these slight differences will be difficult to exploit in the context of fingerprinting in order to identify these devices. Indeed, a fingerprinting algorithm could conclude that these four groups of plots are from the same device and not from four different devices.

As in the previous experiment (Experiment 2), the profile of the PSD was again more sensitive to the changes in data than its amplitude. Indeed, the profile of the PSD changed a lot following the change in data modulating the RF carrier; this is the most important information for us in this experiment.

The similarity between 2 PSDs from 2 different devices, for the same data, in the anechoic chamber (Table 6) was greater than between 2 PSDs from the same device, for the same data, but from two different environments (Table 4).

## 7. Discussion

This survey highlighted the weaknesses of PSD as a fingerprinting feature. The stability under any circumstance of the PSD’s two most important parameters, amplitude and profile, for a given device, is essential to ensure the use of the PSD as a fingerprinting feature. However, our experiments pointed out the limits of the robustness of these two parameters to the environmental conditions and to data modulating the RF carrier. We have shown that the PSD is not at all resilient to changes in either environmental parameters or data modulating the RF carrier.

As expected from a theoretical point of view, we found that the amplitude of the PSD was a quantification of the amount of RF power arriving at the receiving antenna. The higher this power is, the higher the amplitude of the PSD and vice versa. Therefore, all parameters affecting this power also affect the amplitude of the PSD, and among these influencing parameters are the environmental ones such as free-space-loss attenuation, polarization mismatch, multipath and fading effects, interferences, and so on. In addition, the operating temperature can influence the hardware stability (in terms of frequency and RF power shift) on both the transmitter and receiver side.

That is why, to answer our question of whether or not the PSD would be a relevant fingerprinting feature, it is not necessary to perform our experiment for different distances between the transmitter (i.e., the BLE device) and the receiver (i.e., RSA306B). We would just be decreasing the amount of received power by increasing the distance or the other way around; we would be increasing the amount of received power by decreasing the distance. Therefore, the amplitude of the PSD would simply evolve in the same way.

However, the profile of the PSD was less sensitive than its amplitude to the variations in the amount of received power, but below a certain level of this power, even the profile of the PSD started to distort.

When it came to data modulating the RF carrier, their influence on the PSD was more hindering than the influence of the environmental parameters to the use of the PSD as a fingerprinting feature. Indeed, while the environment had more influence on the amplitude of the PSD, the data had more influence on the profile of the PSD, which was more difficult to manage using a fingerprinting algorithm.

In fact, the PSD was signal-specific, the signal changed with the data, and the PSD was specific to the signal; in other words, the PSD varied according to the data, which made it completely useless as a fingerprinting feature in a dynamic IoT environment where the data transmitted by a device can be different from one transmission to another.

Therefore, an environment wherein the PSD could be considered as a relevant feature for fingerprinting would be a static IoT environment with static devices preferably sending unique data, somewhat like a home automation setup. 

Even for such an environment, fingerprinting by PSD is not that efficient; indeed, if it is performed in a centralized way as it was in this work, a simple way to confuse the fingerprinting system would be to get close enough to a device and transmit the same data as this device with the same transmission power.

These experiments, conducted on several devices using BLE technology, showed that a collision is highly probable. Thus, this probability is really not negligible if we consider all the devices manufactured. Then, we showed that device authentication cannot rely on PSD alone, while today’s connected environments can include thousands of devices and there are billions of them already deployed around the world. However, the PSD can be part of the features of an intrusion detection system (IDS). These PSD analyses are particularly interesting, since, as stated in [25], the efficiency of an IDS relies heavily not only on its implementation techniques but also on the relevance of its features and their extraction method. Indeed, relevant features allow increasing the accuracy and reducing the detection time of the IDS. This survey shows that the PSD, if part of an IDS’s features, even in a static connected environment, may introduce a large number of false positives if the comparison is too severe or a large number of false negatives if the comparison is less severe. Features refer to criteria that fingerprinting uses to establish the identity of a transmitter or that an IDS uses to establish the legitimate nature of a communication.

Fingerprinting remains an attractive approach for IoT security, due in part to its genericity, which is one of the most desired characteristics of security solutions for IoT environments. Indeed, fingerprinting is a protocol-independent approach, which makes it suitable for IoT device networks that are sometimes highly heterogeneous with devices using different protocols including proprietary ones. Different frequencies and modulations can also be present in IoT networks.

However, there are still several challenges to overcome, such as finding relevant fingerprinting features, unwanted parameter effects management, as well as fingerprinting solutions cost optimization. Indeed, one of the weaknesses of fingerprinting is that the more similar devices there are in a network, the more likely it is to have some fingerprints so close that it is difficult to disassociate their corresponding devices. For example, the probability of successfully distinguishing several instances of the same device, manufactured in a chain on the same day, will be low compared to cases with completely different devices. This is why fingerprinting requires the use of very precise algorithms and, therefore, very powerful (and, hence, expensive) hardware to increase the efficiency of the approach.

## 8. Conclusions

In this survey, we have highlighted and quantified, through experimental results, the behavior of PSD under the influence of environmental parameters and the changes in data modulating the RF carrier. Our results are proof that the PSD cannot be used alone and without any precaution as a reliable fingerprinting feature in any IoT environment for several reasons.

First, the amplitude of the PSD is sensitive to environmental parameters related to the communication channel and, second, the profile of the PSD is sensitive to data modulating the RF carrier. There are also other parameters (not considered for this study), e.g., device aging, temperature shift, etc., that can impact the PSD over the time. Therefore, the PSD cannot be considered as a relevant fingerprinting feature in a dynamic IoT environment.

Nevertheless, in contrast to the variation in its profile, the variation in the amplitude of the PSD will be easy to manage using a fingerprinting algorithm by using a normalized PSD approach instead of absolute PSD, for example. In fact, the amplitude will not be as important as the profile in identifying a transmitter using the PSD of its emitted signal. Two PSDs with the same profile but different amplitudes are likely from the same device that will have transmitted at two different distances from the fingerprinting receiver. On the other hand, it will be difficult to identify a device using the PSD of its emitted signal if the profile of this PSD changes from one transmission to the next.

The ideal IoT environment for using the PSD as a fingerprinting feature will be an IoT environment where the environmental parameters are not changing, the multipath and fading effects related to propagation channel are limited, the IoT devices are not moving, and each device is always transmitting the same data on the same frequency with the same modulation. However, these conditions, especially the last one, are not quite realistic since even a simple connected smoke detector sends at least two different messages, which are a message when there is smoke and a different message when there is not. The IoT environments meeting these conditions are not numerous, i.e., PSD fingerprinting is not a robust solution for intrusion detection systems suitable for most real-world IoT environments.

Finding solutions to manage the limitation of the fingerprinting approach based on PSD is among the new directions for device fingerprinting research. Some dedicated hardware can be added for fingerprinting purposes on the transmitter and/or receiver side and a dedicated/standardized bit frame (for PSD fingerprinting analysis/identification) can be used in the beginning of a secure (with the fingerprinting approach) wireless communication.

Additionally, the results presented in this paper can be helpful in any study involving PSD, even for applications other than fingerprinting.

## Figures and Tables

**Figure 1 sensors-23-02353-f001:**
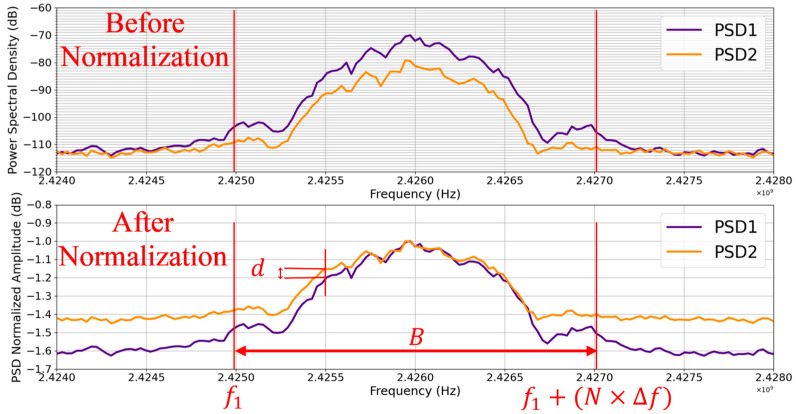
Comparison of different PSDs before and after the normalization. Parameter d is frequency-dependent and is given by$\;d=\left|PSD_{1}^{*}\left(f\right)-PSD_{2}^{*}\left(f\right)\right|$, where * denotes the normalized amplitude. In this figure, d is shown at f=2.4255 GHz as an example.

**Figure 2 sensors-23-02353-f002:**
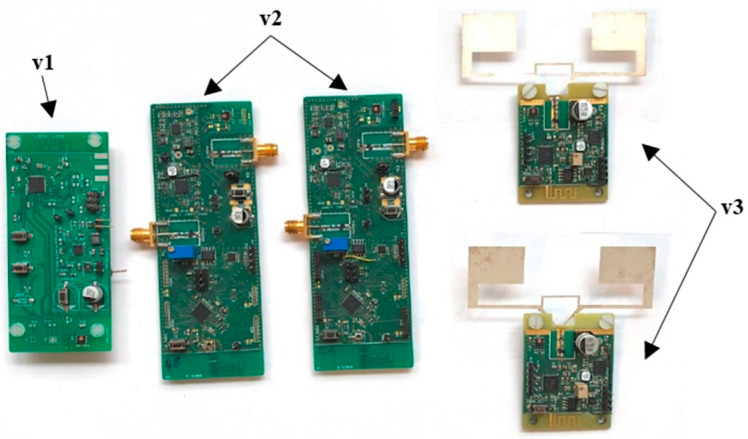
The BLE devices used in our experiments. These devices were designed in the context of the same project; they are different versions of the same product whose hardware and software architectures have evolved over time. Thus, v1 (the board at the left end) refers to the first ever version; v2 (the two boards in the middle) and v3 (the two boards at the right end) refer to the second and third versions, respectively.

**Figure 3 sensors-23-02353-f003:**
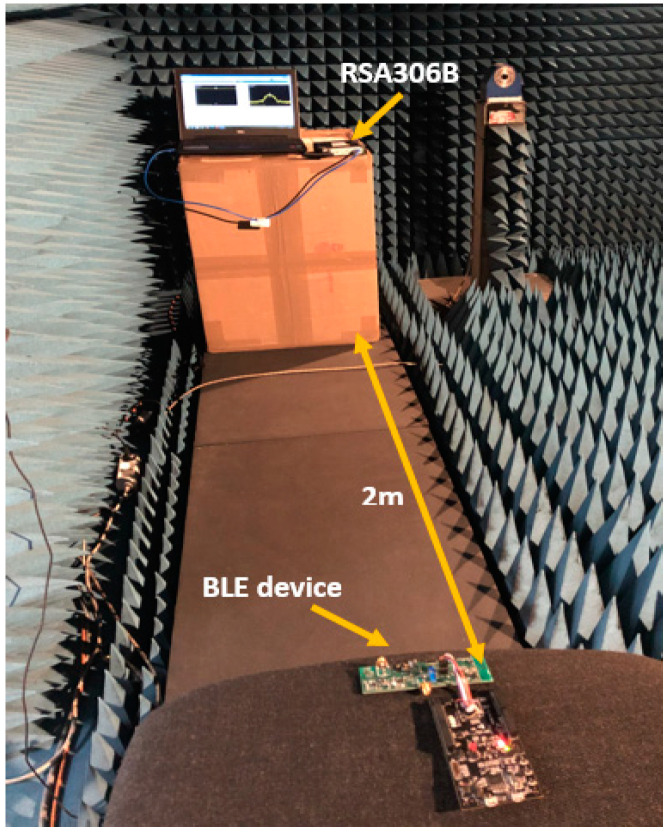
Experimental setup in the anechoic chamber. The BLE device emits the signal at 2 m from the RSA306B. The latter is equipped with a BLE antenna and driven by the Tektronix SignalVu-PC software. This way, we can capture the BLE signal in real time and record it as IQ samples on the PC in order to be used later in the script for estimating the power spectral density using Welch’s average periodogram method.

**Figure 4 sensors-23-02353-f004:**
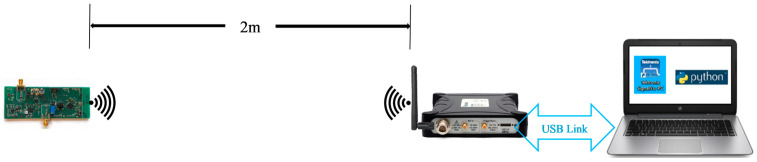
Experiment schematic.

**Figure 5 sensors-23-02353-f005:**
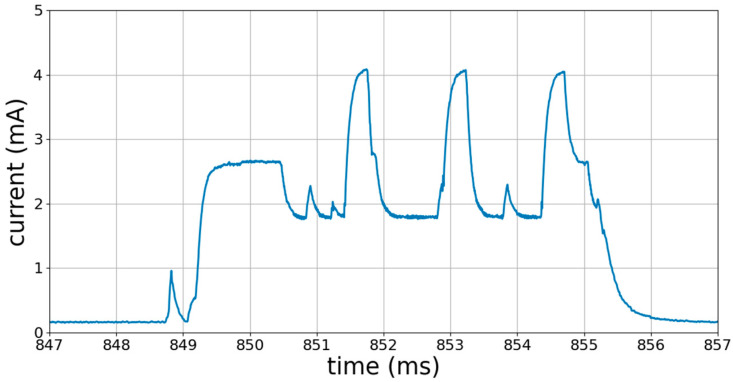
Profile of the power consumption of the BLE device during advertising, i.e., the broadcast of the advertising packet on each of the three primary advertising channels, one after another. Each peak corresponds to the advertising on a channel.

**Figure 6 sensors-23-02353-f006:**
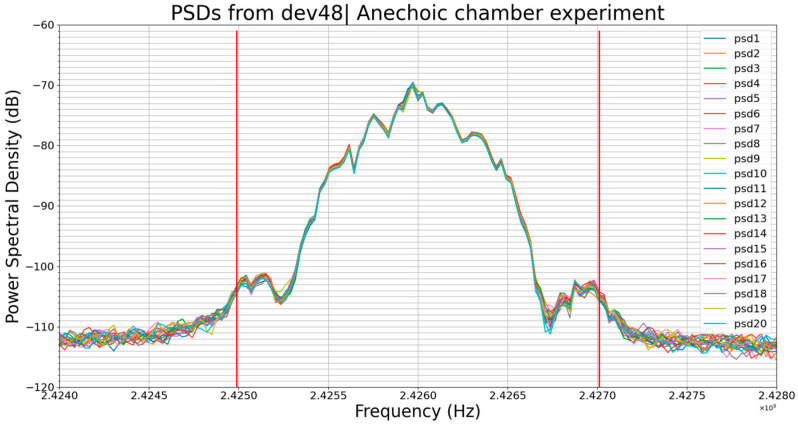
Twenty PSDs of a same BLE device in a static experimental setting. The region between the red lines is the band *B* where the PSDs are compared; we chose *B* = 2 MHz, which is the channel bandwidth of the BLE. So, we ignored the background noise to have a more relevant analysis.

**Figure 7 sensors-23-02353-f007:**
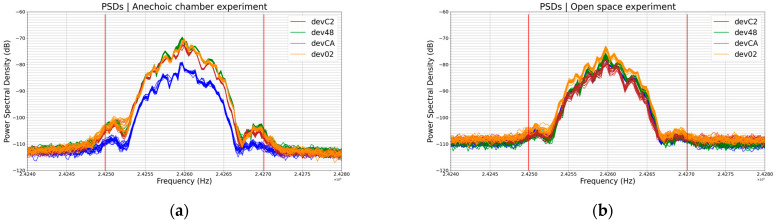
(**a**) PSDs of anechoic chamber experiment (**b**) PSDs of open-space experiment.

**Figure 8 sensors-23-02353-f008:**
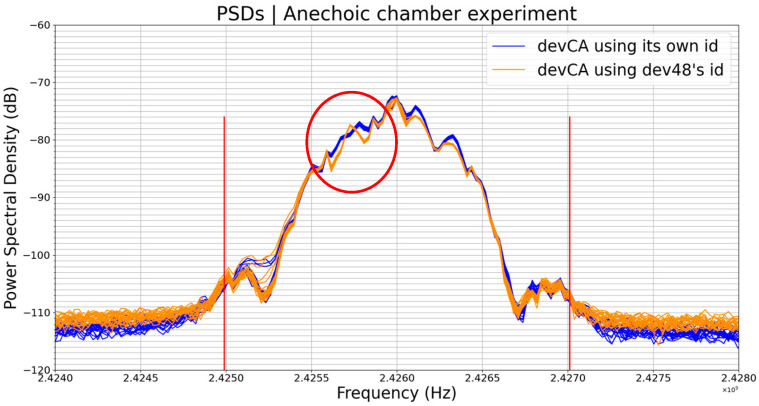
PSDs of the same BLE device but measured with two different identifiers.

**Figure 9 sensors-23-02353-f009:**
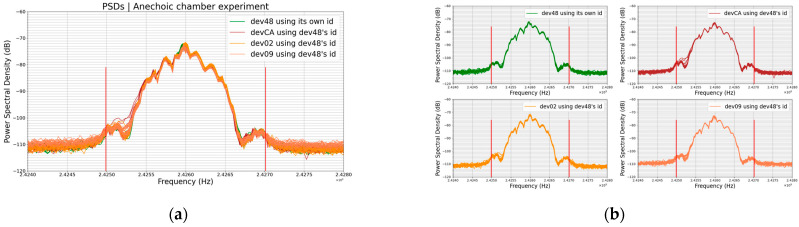
(**a**) PSDs of four different BLE devices using the same identifier (the one of dev48) and, thus, always transmitting exactly the same data. Graph (**a**) plots are an overlapping of graph (**b**) four groups of plots.

**Figure 10 sensors-23-02353-f010:**
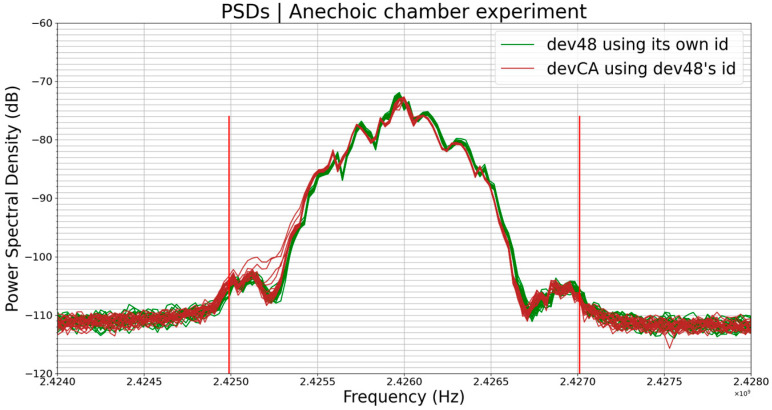
PSDs of dev48 (green plots) and devCA (brown plots) devices using the same identifier (the one of dev48) and, thus, always transmitting exactly the same data.

**Table 1 sensors-23-02353-t001:** Summary of the experiments in this study.

	Experiment Parameters	Experiment Name	Experiment Environment	BLE Device Name	BLE Device Identifier	Distance between BLE Device and Receiver	Number of PSDs (i.e., Number of Transmission)
**Experiment** **Goal**							
Assess the influence of environmental parameters on the PSD.		Experiment 1	Anechoic chamber	devC2	E0512496A1C2	2 m	20
				dev48	E05124969648	2 m	20
				devCA	E0512496C3CA	2 m	20
				dev02	B87C6FCD6102	2 m	20
			Open space	devC2	E0512496A1C2	2 m	20
				dev48	E05124969648	2 m	20
				devCA	E0512496C3CA	2 m	20
				dev02	B87C6FCD6102	2 m	20
Assess the influence of the data modulating BLE RF carrier on the PSD.		Experiment 2	Anechoic chamber	devCA	E0512496C3CA	2 m	20
				devCA	E05124969648	2 m	20
		Experiment 3	Anechoic chamber	dev48	E05124969648	2 m	20
				devCA	E05124969648	2 m	20
				dev02	E05124969648	2 m	20
				dev09	E05124969648	2 m	20

In “Experiment 1”, our objective was to compare the PSDs in anechoic chamber with those in open space. In “Experiment 2”, we had the same device, devCA, which transmitted in first case with its own identifier; then, in second case, with the identifier of dev48, we compared the PSDs of these two cases. In “Experiment 3”, the three devices devCA, dev02, and dev09 all used dev48’s identifier, so these four devices transmitted the same data. Could we distinguish their PSDs? This question is answered in “Section 6.2” below.

**Table 2 sensors-23-02353-t002:** Configuration parameters of the RSA306B real-time spectrum analyzer.

Parameter	Value	Comments
Frequency	2426 GHz	Second advertising channel.
Span	4 MHz	This is two times the BLE channel bandwidth.
Sampling frequency	56 MS/s	
Acquisition Length	1 ms	This is 56k samples.
Acquisition start-up	Triggered on RF input	Trigger level = −55 dBm.
		Slope = Rise.
Save acq data on Trigger	Checked	File format selected = CSV
Ref Lev	−33 dBm	Internal preamplifier on.

The “Trigger Level” value depends on the power of the received signal; in our case, this power was very low. Fortunately, the RSA306B had an internal preamplifier which was automatically activated when the value of the parameter “Ref Level” was lower than or equal to −30 dBm.

**Table 3 sensors-23-02353-t003:** Results of the comparison of the 20 PSDs.

Pair of PSDs	*ε*(dB) = 0.025		*ε*(dB) = 0.05	
	*N_F_*	*S_p_* (%)	*N_F_*	*S_p_* (%)
psd1 vs. psd2	69	92.00	69	92.00
psd3 vs. psd4	67	89.33	70	93.33
psd5 vs. psd6	70	93.33	70	93.33
psd7 vs. psd8	71	94.66	71	94.66
psd9 vs. psd10	67	89.33	70	93.33
psd11 vs. psd12	73	97.33	74	98.66
psd13 vs. psd14	71	94.66	71	94.66
psd15 vs. psd16	70	93.33	70	93.33
psd17 vs. psd18	70	93.33	73	97.33
psd19 vs. psd20	67	89.33	68	90.66

**Table 4 sensors-23-02353-t004:** Results of the comparison of the average PSDs of anechoic chamber experiments with the ones of open-space experiments for each device.

BLE Device Name	*ε*(dB) = 0.025		*ε*(dB) = 0.05	
	*N_F_*	*S_p_* (%)	*N_F_*	*S_p_* (%)
devC2	65	86.66	65	86.66
dev48	39	52.00	46	61.33
devCA	32	42.66	40	53.33
dev02	31	41.33	56	74.66

The parameters B and Δf keep the same values as in Section 6.2.1.

**Table 5 sensors-23-02353-t005:** Results of the comparison of the average PSDs of the same device transmitting two different identifiers.

BLE Device Name	*ε*(dB) = 0.025		*ε*(dB) = 0.05	
	*N_F_*	*S_p_* (%)	*N_F_*	*S_p_* (%)
devCA	64	85.33	68	90.66

The parameters B and Δf keep the same values as in Section 6.2.1.

**Table 6 sensors-23-02353-t006:** Results of the comparison of the average PSDs of four devices transmitting the same data.

Pair of PSDs	*ε*(dB) = 0.025		*ε*(dB) = 0.05	
	*N_F_*	*S_p_* (%)	*N_F_*	*S_p_* (%)
dev48 vs. devCA	38	50.66	51	68.00
dev48 vs. dev02	67	89.33	68	90.66
dev48 vs. dev09	58	77.33	62	82.66
devCA vs. dev02	41	54.66	54	72.00
devCA vs. dev09	57	76.00	64	85.33
dev02 vs. dev09	57	76.00	63	84.00

The parameters B and Δf keep the same values as in Section 6.2.1.

## Data Availability

Section 6.1 provided required information to reproduce experiments. We still can provide our data on request.

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
