# Peer review of "Intrusion Detection System for IoT: Analysis of PSD Robustness"

_sensors, 2023, doi:10.3390/s23042353_

Round 1
Reviewer 1 Report
The paper presents to known whether or not the Power Spectral Density (PSD) of the RF signal from a 20 device could be a relevant feature for its fingerprinting.
The authors have done a credible job of pulling together a number of disparate sets of activities into a single manuscript.
This is both a strength and a weakness of the paper, but in this referee’s opinion, the balance is strongly on the “strength” side.
Given the large variety of data sources, they did a fine job keeping focus and consolidating where possible.
I do have a few suggestions to improve the paper for the reader, which I will detail below.
With the major suggested improvements, I expect the manuscript will be publishable.
I suggest is tackle the numerous differences in the data sources in a much more head on fashion.
The authors do a very good job of describing the differences in measurement methods in Section 2.
What is missing in each case is a discussion of the significance of the differences. Are the differences important?
Do they affect the results in any way? How might one evaluate the effects of these differences in the overall study.
I recommend rewriting the introduction.
The literature could be improved and below references are advised to be read to improve the state of art:
Fire resistance prediction of slim-floor asymmetric steel beams using single hidden layer ANN models that employ multiple activation functions.
Higher circulating levels of chemokines CXCL10, CCL20 and CCL22 in patients with ischemic heart disease.
Application of a grounded group decision-making (GGDM) model: a case of micro-organism optimal inoculation method in biological self-healing concrete.
I would like to have detailed explanations for The key challenges of intrusion detection and how to overcome them including:
ensuring an effective deployment.
knowing how to respond to threats.
managing the high volume of alerts.
understanding and investigating alerts.
Author Response
Please find attached our feedbacks to your valuable suggestions and comments.

Reviewer 2 Report
The researcher investigated whether the power spectral density (PSD) of the RF signal from the device could be a relevant feature of its signature. He assured him that the relevant fingerprint feature should remain stable regardless of environmental conditions, over time and under the influence of any other parameters,
He performed experiments in which he was able to identify the limits and possibilities of power spectral density (PSD) as a fingerprint feature. Thus, the researcher overcame one of the defects of the devices, and thus he could overcome the reflection of the radio frequency signal of a wireless communicating device
I suggest accepting the search
Author Response

(The authors gave the same response as above.)

Reviewer 3 Report
1- author need to clearly state the contributions of the study in introduction section
2- all equations, figures, and tables should mentioned in body text before using them,
3- add numerical results to conclusion
4- enrich the work by citing recent published work such as
https://mesopotamian.press/journals/index.php/CyberSecurity/article/view/8
https://mesopotamian.press/journals/index.php/CyberSecurity/article/view/1
https://mesopotamian.press/journals/index.php/CyberSecurity/article/view/7
Author Response

(The authors gave the same response as above.)

Round 2
Reviewer 1 Report
Concerning the initial round of questions addressing discrepancies in data sources:
Can you elaborate on how you've dealt with the various data sources?
Have you examined the relevance of these distinctions and how they might influence the results?
Could you clarify how the reader might assess the effect of these variations on the overall study?
With respect to the revision of the introduction:
Can you elaborate on the changes you made to the introduction?
Have you established consistency with the work's objectives while clearly stating the study's contribution and position?
With respect to the literature review:
Can you clarify why the listed sources do not directly pertain to the current study?
Have you examined any further sources that might advance the state of the art?
Regarding the description of significant obstacles in intrusion detection:
Can you elaborate on how to overcome the most significant obstacles in intrusion detection, such as assuring successful deployment, responding to attacks, handling a high amount of alerts, and investigating alerts?
How does PSD fit into the overall security solution for fingerprinting and intrusion detection systems?
Regarding the English language's editing:
Can you describe the modifications made to the manuscript's English language and style?
Have you verified that the amended text adheres to vocabulary and grammar rules?
Author Response
Please see the attached file.
Kind regards,
Lamoussa SANOGO and Daniela DRAGOMIRESCU
